# The Impact of Repeating COVID-19 Rapid Antigen Tests on Prevalence Boundary Performance and Missed Diagnoses

**DOI:** 10.3390/diagnostics13203223

**Published:** 2023-10-16

**Authors:** Gerald J. Kost

**Affiliations:** 1Pathology and Laboratory Medicine, School of Medicine, University of California, Davis, CA 95616, USA; geraldkost@gmail.com; 2Point-of-Care Testing Center for Teaching and Research (POCT•CTR), Knowledge Optimization, Davis, CA 95616, USA

**Keywords:** coronavirus disease-2019 (COVID-19), emergency use authorization (EUA), false negative (FN), false omission rate (R_FO_), point-of-care testing (POCT), prevalence boundary (PB), rapid antigen test (RAgT), repeated testing, sensitivity and specificity, tier

## Abstract

A prevalence boundary (PB) marks the point in prevalence in which the false omission rate, R_FO_ = FN/(TN + FN), exceeds the tolerance limit for missed diagnoses. The objectives were to mathematically analyze rapid antigen test (RAgT) performance, determine why PBs are breeched, and evaluate the merits of testing three times over five days, now required by the US Food and Drug Administration for asymptomatic persons. Equations were derived to compare test performance patterns, calculate PBs, and perform recursive computations. An independent July 2023 FDA–NIH–university–commercial evaluation of RAgTs provided performance data used in theoretical calculations. Tiered sensitivity/specificity comprise the following: tier (1) 90%, 95%; tier (2) 95%, 97.5%; and tier (3) 100%, ≥99%. Repeating a T2 test improves the PB from 44.6% to 95.2% (R_FO_ 5%). In the FDA–NIH-university–commercial evaluation, RAgTs generated a sensitivity of 34.4%, which improved to 55.3% when repeated, and then improved to 68.5% with the third test. With R_FO_ = 5%, PBs are 7.37/10.46/14.22%, respectively. PB analysis suggests that RAgTs should achieve a clinically proven sensitivity of 91.0–91.4%. When prevalence exceeds PBs, missed diagnoses can perpetuate virus transmission. Repeating low-sensitivity RAgTs delays diagnosis. In homes, high-risk settings, and hotspots, PB breaches may prolong contagion, defeat mitigation, facilitate new variants, and transform outbreaks into endemic disease. Molecular diagnostics can help avoid these potential vicious cycles.

## 1. Introduction

The high specificity of Coronavirus disease-19 (COVID-19) rapid antigen tests (RAgTs) helps minimize false positives, although at very low prevalence (e.g., <2%), they may appear [1,2,3,4,5,6,7,8,9,10,11]. However, RAgTs fail to reliably rule out infections because poor clinical sensitivity produces false-negative results [12,13,14]. The prevalence boundary (PB) is defined as the prevalence at which the rate of false omissions, R_FO_ = FN/(TN + FN), exceeds a specified threshold, such as 5% or 1 in 20 diagnoses missed because of false negatives (FNs).

The objectives of this research are to mathematically reveal patterns of RAgT performance, to understand intrinsic limitations imposed on RAgT performance, to determine why and where PBs are breeched, and to evaluate the merits of repeating RAgTs twice at intervals of 48 h over 5 days for a total of three tests, which is the temporal protocol now required by the US Food and Drug Administration (FDA) for people who are asymptomatic. The overall goal is to develop a sound mathematical basis for improving RAgTs and designing new ones well in advance of the next pandemic.

## 2. Methods

### 2.1. Viewpoint

The viewpoint here is post hoc Bayesian conditional probability, that is, the perspective of the healthcare provider or self-testing layperson who must judge whether a positive COVID-19 test result is believable, and likewise, decide whether or not to trust a negative test result to rule out infection. For quantitative 2 × 2 tables illustrating how prevalence affects test performance and generates false negatives, please see Tables 3–6 in reference [1].

### 2.2. Performance Tiers and Mathematical Foundations

Table 1 presents the performance tiers and quantitates the effects of repeating tests on prevalence boundaries when R_FO_ is 5% [1,13,14]. Tier 2 performance produces a PB of 95.2% upon repeating a test, which is numerically about the same as the sensitivity of 95%. A tier 3 test has clinical sensitivity [TP/(TP + FN)] of 100% and thus has no FNs. It does not need to be repeated unless it is useful to confirm positive test results.

Table 2 lists the equations, dependent variables, and independent variables used to graph R_FO_ versus prevalence [Equation (21)] and the gain in PB (∆PB) [Equation (26)] following repetition of a test versus its sensitivity. Please note that the righthand side of Equation (26) for ∆PB *does not depend on prevalence*, per se. A graph based on Equation (26) shows how ∆PB changes as a function of sensitivity.

Table 2 also lists the equations used to determine the R_FO_ for a repeated test (R_FO/rt_) [Equation (22)], the PB for one test given the R_FO_ [Equation (24)], and the PB for a repeated test (PB_rt_) given the R_FO_ [Equation (25)]. Equations (22), (25), and (26) were newly derived and verified for this research.

### 2.3. Prevalence Boundaries

A prevalence boundary is encountered where the R_FO_ curve (plotted as a function of prevalence) intersects the threshold for missed diagnoses. This paper uses a primary threshold of 5%, but it also illustrates the effects of R_FO_ thresholds of 10%, 20%, and 33%. Equation (21) is used to calculate the R_FO_, and Equation (22) is used to calculate the R_FO/rt_ when a test is repeated. Please note that ∆PB [Equation (26)], the gain in PB from repeating a test, depends only on the sensitivity, specificity, and R_FO_.

### 2.4. FDA, NIH, University, and Commercial RAgT Field Evaluation (the “Collaborative Study”)

Rapid antigen test results for an asymptomatic “DPIPP 0–6” group in a collaborative study of COVID-19 diagnosis conducted by Soni et al. were published in a preprint in 2022 [15] and in July 2023 in a peer-reviewed journal [16]. The study was conducted from 18 October 2021 to 31 January 2022. It included subjects over two years of age and was funded by the Rapid Acceleration and Diagnostics (RADx) initiative of the NIH.

The collaborative study involved 7361 participants with 5609 deemed eligible for analysis in the preprint [15] and 5353 in the peer-reviewed paper [16]. Participants who were symptomatic and negative for SARS-CoV-2 on study day one were eligible. In total, 154 participants had at least one positive RT-PCR result. The collaborative study was approved by the WIRB-Copernicus Group Institutional Review Board (20214875) [16].

Participants were eligible to enroll through a smartphone app if they had not had a SARS-CoV-2 infection in the prior three months, had been without any symptoms in the fourteen days before enrollment, and were able to drop off prepaid envelopes with nasal swab samples at their local FedEx drop-off location.

People self-tested and self-interpreted RAgT results during the spread of SARS-CoV-2 Delta and Omicron variants. The results, which reflected the first week of testing, were used here to analyze repeated RAgT performance. Soni et al. [16] pooled RAgT performance results across the tests used while assuming similar sensitivity for viral loads and thereby created an adequate sample size to fulfill the goals of the FDA–NIH–university–commercial consortium.

Institutions that participated in the collaborative study consortium comprised the US FDA, National Institute of Biomedical Imaging and Bioengineering at the NIH, University of Massachusetts Chan Medical School, Johns Hopkins School of Medicine, and Northwestern University. Quest Diagnostics and CareEvolution also joined the evaluation.

### 2.5. Rapid Antigen Tests in the Collaborative Study

The RAgTs used in the collaborative study under FDA emergency use authorizations [17] (EUAs) included the (a) Abbott BinaxNOW Antigen Self Test [EUA positive percent agreement (PPA), 84.6%; negative percent agreement (NPA), 98.5%]; (b) Quidel QuickVue OTC COVID-19 Test [PPA, 83.5%; NPA, 99.2%]; and (c) BD Veritor At-Home COVID-19 Test [PPA, 84.6%; NPA, 99.8%]. The EUA NPA range from 98.5 to 99.8% has a narrow span of only 1.3%.

The collaborative study preprint [15] did not report clinical specificity results. The peer-reviewed paper stated that 3.4% (1182) of same-day RT-PCR negative results “were missing a corresponding Ag-RDT result” [16]. Then, the authors estimated the clinical specificity [TN/(TN + FP)] to be 99.6%. Rather than using an estimated specificity, the median EUA NPA of 99.2% was used here for mathematical analyses.

In a study of COVID-19 test performance [18], the median NPA of EUA manufacturer claims for home RAgTs was 99.25% (range 97–100%), which is nearly identical to the 99.2% used here for math computations. For commercial EUA NPA details, please see “Table S1, Part I. Antigen tests, Statistics” [19] in the supplement to reference [18].

### 2.6. FDA Directive for Rapid Antigen Tests

The collaborative study preprint [15] was followed by a letter from the US FDA titled “Revisions Related to Serial (Repeat) Testing for the EUAs (Emergency Use Authorizations) of Antigen IVDs” [20], published 1 November 2022. Appendix A in the FDA letter states “(1) Where a test was previously authorized for testing of symptomatic individuals (e.g., within the first [*number specific to each test*] days of symptom onset), the test is now authorized for use at least twice over three days with at least 48 h between tests.”, and “(2) Where a test was previously authorized for testing of asymptomatic individuals (e.g., individuals without symptoms or other epidemiological reasons to suspect COVID-19), the test is now authorized for use at least three times over five days with at least 48 h between tests”.

Intended use EUA documents describing RAgTs must now declare that “negative results are presumptive”, and no longer specify that testing should be performed at least twice over 2–3 days with 24–36 h between tests. Product labeling must be updated along with instructions for users and other manufacturer documents. This research focuses on the interpretation of test results for asymptomatic subjects (FDA no. 2 above).

### 2.7. Software and Computational Design

Desmos Graphing Calculator v1.9 [https://www.desmos.com/calculator (accessed on 5 September 2023)], which is a free multivariate open access software, was used to generate illustrations so that readers could duplicate the graphical results and explore their analytic goals at no expense. Mathematica [Wolfram, https://www.wolfram.com/mathematica/ (accessed on 5 September 2023), ver. 13.3] was used to confirm the (x, y) coordinates of graphical intersections and other analytical results.

### 2.8. Human Subjects

Human subjects were not involved in the mathematical analyses. Sensitivity and specificity data used here were obtained from public-domain-published sources [15,16].

## 3. Results

Figure 1 shows the changes in the false omission rates, R_FO_, as a function of prevalence from 0 to 100%. The red curves reflect the results of testing three times for asymptomatic self-testers participating in the collaborative study. Please see the inset table for details.

Repeating RAgTs improved sensitivity from an initial 34.4% to 55.3% on the first repetition and 68.5% on the second repetition when singleton RT-PCR positives were included. The initial PB was 7.37% (red dot). However, subsequent PBs (10.46%, 2nd test; 14.22%, 3rd test) did not match the theoretical predictions for repetitions. 

In community settings and hotspots with prevalence >7.37%, the R_FO_ curves predict that more than 1 in 20 diagnoses will be missed with the first test, while with the second and third tests, R_FO_ breeches will occur at 10.46% and 14.22% prevalence, respectively.

Relaxation of the R_FO_ threshold to 10%, 20%, and 33.3% for the third test generates unacceptable levels of missed diagnoses (1 in 10, 1 in 5, and 1 in 3, respectively) as the PB moves up and to the right at 25.9, 44.1, and 61.2% prevalence, indicated by the red symbols on the exponentially increasing red curve for the third test.

The second repetition of the RAgTs (third RAgT) did not achieve the World Health Organization (WHO) performance criteria [21] (blue dot and curve, Figure 1) for RAgT sensitivity of 80% and generated a PB of only 14.22% (R_FO_ = 5%), which is 69.9% of the PB (20.34%) calculated (using Equation (24)) for the WHO specifications.

The highest levels of performance in Figure 1 were attained by the single home molecular loop-mediated isothermal amplification (LAMP, purple curve) assay median performance [22] (sensitivity 91.7%, specificity 98.4%, and PB 38.42%), the mathematically predicted performance of a tier 2 test (PB 50.6%, green curve), and the tier 2 repeated test (PB 95.2%, large green dot). Tier 2 sensitivity is 95%; for R_FO_ = 5%, the predicted PB would increase by 44.6% to 95.2% when the test is repeated.

Figure 2 displays gain in the prevalence boundary, ∆PB, on the vertical (y) axis versus the sensitivity of the test on the horizontal (x) axis. Initially, the ∆PB curve is relatively shallow. As sensitivity increases, ∆PB peaks at 91.0 to 91.4% (see the magnifier at the top). The curves cluster together because of the small span in specificity (see the left column of the inset table). The magnifier at 25% ∆PB shows that the relative order within the cluster is the same as the ranking by specificity in the inset table.

The righthand columns of the inset table in Figure 2 list actual PBs and theoretical predictions. For the collaborative study, the gain in PB obtained with the first repeated test, 3.09%, approximated that predicted, 3.37%. Upon testing twice, the gain in PB of 3.76% was only 37.1% of the 10.13% predicted. There is no clear explanation for the meager improvement.

The PBs for the second and third tests, 10.46% and 14.22%, respectively, lagged behind the theoretical predictions of 10.74% and 20.59%, respectively. The two red boxes show where the repetition points lie on the red ∆PB curve and explain the progression of PBs. The arrows point to the coordinates of ∆PB (y axis) and sensitivity (x axis).

Looking back at Figure 1, we see that for R_FO_ = 5%, the median of home molecular diagnostic LAMP tests (HMDx, purple curve) performs better with just one test than three serial RAgTs and beats WHO performance by positioning itself between the WHO and tier 2 R_FO_ curves. In general, the plot of ∆PB versus sensitivity in Figure 2 reveals that when one tolerates 1 in 20 missed diagnoses, repeating a test will not increase the PB maximally unless the sensitivity is 91.03–91.41%.

In Figure 2, the curves cluster together (see magnifiers) in the right-skewed peak shape because specificity is uniformly high (95–99.2%). The rate of gain in ∆PB depends primarily on sensitivity (x axis) and follows the slope of the curve cluster. The slope is highest from about 75–85%, which implies test performance has the most to gain there. 

## 4. Discussion

Clinical evaluations show that the specificity of COVID-19 RAgTs is high [1,2,3,4,5,6,7,8,9,10,11,18,19]. In Figure 2, the ∆PB curves cluster together because the range of specificity (95–99.2%) is narrow. Therefore, the degree to which a repeated RAgT increases the PB depends primarily on the test sensitivity. This mathematical analysis is not exclusive to COVID-19 testing. It applies to other positive/negative qualitative diagnostic tests for infectious diseases and can help optimize future assay designs.

Investigators have addressed the sensitivity of RAgTs in various settings. In hospitalized patients, Kweon et al. [23] found that for RT-PCR cycle thresholds of 25–30, point-of-care antigen test sensitivity ranged from 34.0% to 64.4%, with higher sensitivity within the first week. Hirotsu et al. [24] reported that antigen testing exhibited 55.2% sensitivity and 99.6% specificity in 82 nasopharyngeal specimens from seven hospitalized patients tested serially. Veroniki et al. [25] documented sensitivity of 55% in studies of asymptomatic subjects. Gallardo-Alfaro et al. [26] found RAgT sensitivity of 50% in asymptomatic children.

In twenty community clinical evaluations of asymptomatic subjects, RAgT sensitivity ranged from 37% to 88% (median 55.75%), and specificity ranged from 97.8% to 100% (median 99.70%) [19]. During a nursing home outbreak, Mckay et al. [27] documented a RAgT sensitivity of 52% with asymptomatic patients. For home RAg testing, Chen et al. [28] reported a negative predictive rate of 38.7% in children. With daily testing, Winnnett et al. [29] observed a clinical sensitivity of 44% and concluded that RAgTs miss infected and presumably infectious people.

In correctional facilities, Lind et al. [30] showed that serial RAgTs had higher but diminishingly improved sensitivities over time, similar to the diminishing returns seen with repeat testing in the collaborative study. In a university setting, Smith et al. [31] found that serial testing multiple times per week increased the sensitivity of RAgTs. Wide variations in sensitivity in these studies and others indicate that for RAgTs to rule out disease, performance should be improved and more consistent with less uncertainty [13].

Asymptomatic infections highlight the need to moderate false negatives, that is, curtail missed diagnoses and assure that repeating RAgTs shifts PBs to the right to mitigate the spread of disease. Soni et al. [32] showed that 31.3% were asymptomatic in a clinical study of serial RAgTs. When comparing RAg to RT-PCR positive test results, Sabat et al. [33] found that 59.5% were asymptomatic. In fifteen studies reviewed, Gao et al. [34] concluded that asymptomatic patients had a significantly lower (27.1%) positivity rate than symptomatic patients (68.1%) on day five.

The schematic in Figure 3 illustrates how missed diagnoses might trigger dysfunctional outcomes depending on the increase in local prevalence, the timing of testing, and the pattern of infectivity. Starting in the top left, highly specific tests may generate false positives when prevalence is very low (e.g., <2%) [1]. For graphs of false positive to true positive ratios versus prevalence, please see Figure 1 in reference [1].

Patients with false-positive COVID-19 test results will generally be isolated (upper left, Figure 3) and cannot spread disease because they are not infected with SARS-CoV-2. The prevalence in the collaborative study was in the range of 2.39 to 2.75% (134/5609 to 154/5609) in late 2021 and early 2022 when data were collected [15]. The singleton RT-PCR positives reported by the investigators may have been false-positive RT-PCR reference test results; to avoid bias in the present study, singletons were not excluded. 

As prevalence increases, the weighting of RAgT performance shifts from specificity to sensitivity (top sequences in Figure 3). A vicious cycle may develop as diagnoses are missed. Repeating low-sensitivity RAgTs does not advance PBs substantially (see Figure 2). False negatives will increase exponentially (see Figure 1) as prevalence hits double digits. Pollán et al. [35] reported seroprevalence > 10% in Madrid in 2020. Gomez-Ochoa et al. [36] reported healthcare worker prevalence of 11% with 40% asymptomatic. 

In 2020, Kalish et al. [37] documented 4.8 undiagnosed infections for every case of COVID-19 in the United States. The 2022 meta-analysis of Dzinamarira et al. [38] found 11% prevalence of COVID-19 among healthcare workers. In a 2021 meta-analysis, Ma et al. [39] discovered that asymptomatic infections were common among COVID-19 confirmed cases, specifically 40.5% overall, 47.5% in nursing home residents or staff, 52.9% in air or cruise travelers, and 54.1% in pregnant women.

Prevalence can be estimated from positivity rates using Equation (30) when high-sensitivity RT-PR testing is used. For example, if the positivity rate is 5%, sensitivity is 100% and specificity is 99% (Tier 3), estimated prevalence will be ~4%, and if the positivity rate is 20% and the specificity is 97.5% with 100% sensitivity, then estimated prevalence will be~18%. Cox-Ganser et al. [40] documented test positivity percentages of up to 28.6% in high-risk occupations. In 2020, the median New York City positivity was 43.6% (range 38–48.1 across zip codes) [41]; estimated prevalence would be 43.0%.

Thus, RAgTs and other COVID-19 diagnostic tests must perform well over wide ranges of contagion that varies geographically, in time, and biologically. For example, Golden et al. [42] showed that antigen concentrations are related to viral load; the limit of detection predicts test performance. Higher-sensitivity point-of-care molecular diagnostics (left in Figure 3), such as LAMP assays [22] with EUAs for home testing or other portable molecular diagnostics, offer a way out of the vicious cycle. Exiting the vicious cycle with highly sensitive and highly specific molecular testing will decrease community risk and enhance resilience [43], including now, as new waves of COVID-19 appear.

The Eris variant, EG.5 (a descendent lineage of XBB.1.9.2), and new BA.2.86 currently threaten well-being, especially that of the elderly. Time spent testing is important. Delaying diagnosis increases the risk of infecting close contacts (see the inner feedback loop in Figure 3). Asymptomatic people carrying SARS-CoV-2 may unknowingly spread the disease to family, friends, workers, and patients as viral loads increase during the protracted three-test, 5-day protocol now mandated by the US FDA for RAgTs. Delays allow new variants to emerge, which in turn increase prevalence.

The US FDA now requires RAgT labeling to state that results are “presumptive”. RT-PCR or other COVID-19 molecular diagnostic tests should be used to confirm negative RAgT results. The WHO and the US declared an end to the pandemic, but people still need to test [44,45]. For the week ending 29 July, 9056 new US hospitalizations were reported, ER cases doubled, and the positivity rate rose to 8.9% for tests reported to the CDC [46]. By September, the positivity rate was over 16% in some regions of the United States [47].

There are limitations to this work. First, Bayesian theory was not proven during the pandemic, although it appears to adequately explain testing phenomena. Second, self-testing in the collaborative study was not controlled, the reference test comparison was incomplete, home QC was omitted, and reagents may have degraded. Third, the layperson testing technique may have been faulty or inconsistent. Fourth, manufacture PPA and NPA specifications may have been overstated in the small studies submitted to the FDA to obtain EUAs.

Further, there was no comparison LAMP molecular assay included in the collaborative study for parallel self-testing at points of care. Nonetheless, these limitations do not obviate the need for higher performance standards and the upgrading of RAgT and other diagnostic assays that will be needed for future surges and threats. Timely diagnosis of COVID-19 is important, especially for children this fall. Mellou et al. [48] found that 36% of children who self-tested were asymptomatic, the median lag to testing positivity was two days, and early diagnosis “…probably decreased transmission of the virus…”.

## 5. Conclusions and Recommendations

Speed and convenience are two of the primary reasons people seek COVID-19 self-testing [18,22]. Repeating RAgTs three times over five days defeats the purpose of *rapid* point-of-care testing, does not inform public health in a timely manner, could complicate contact tracing, and may not be cost-effective.

Missed diagnoses can perpetuate virus transmission, exponentially more so when prevalence exceeds PBs. Tolerances limits for missed diagnoses have not been established, nor have they been tied to different levels of prevalence. The ∆PB [Equation (26)] does not depend on prevalence and should be optimized if tests are repeated by using those with very high sensitivity (i.e., tier 2 or tier 3).

No precise temporal trend maps of COVID-19 prevalence in different countries are available for comparison, so the impact of prevalence, per se, is uncertain, although prevalence is known to have been very high in COVID-19 hotspots and high-risk settings [49]. Breaches of RAgT PBs may have generated vicious cycles, adversely transformed outbreaks into endemic disease, prolonged contagion, defeated mitigation, allowed new variants to arise, and fueled the pandemic, as Figure 3 and the prevalence boundary hypothesis [43] suggest.

The FDA allowed manufacturers to support RAgT serial screening claims with new clinical evaluations [20]. Upgraded performance should be demonstrated in multicenter trials with large numbers of diverse subjects. To decrease missed diagnoses with a repeated test, mathematical analysis suggests that RAgT sensitivity should be 91.03 to 91.41% in actual clinical evaluations. The theory also shows that a test with a tier 2 clinical sensitivity of 95% will generate PB of 95.2% when only repeated once (see Table 2).

Use of RAgTs for COVID-19 or future highly infectious disease threats should be evidence-based [49]. COVID-19 was shown to have positivity rates and/or prevalence as high as 75% in California and Ohio prisons and in emergency rooms in Brooklyn, New York [50], which creates high potential for asymptomatic infections to spread silently. If superior RAgT performance is not attainable, the FDA should retire EUAs. Rapid antigen tests should achieve performance levels proven clinically to be at least tier 2 (95% sensitivity, 97.5% specificity), especially in high-risk settings and infectious disease hotspots.

## Figures and Tables

**Figure 1 diagnostics-13-03223-f001:**
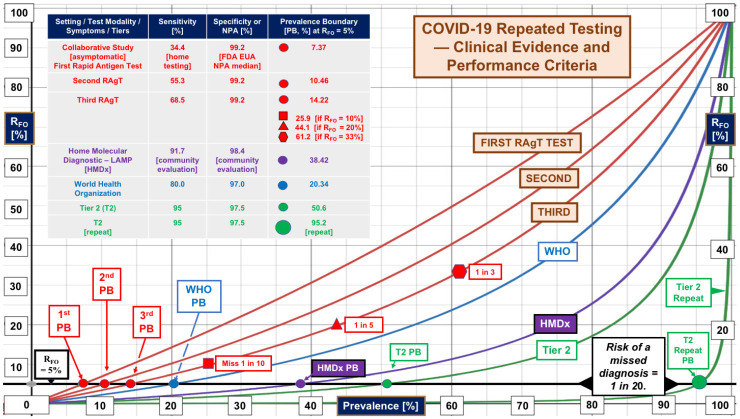
**False Omission Rates Increase Exponentially with Prevalence**. The median performance of a home molecular diagnostic test (HMDx LAMP, purple curve) performed only once beats that for three serial RAgTs in the collaborative study. A repeated tier 2 test (green curve rising on the right) will not miss more than 1 in 500 diagnoses until the prevalence exceeds 43.8%, then 1 in 200 up to 65.9% prevalence, and subsequently 1 in 100 up to 79.4%, 1 in 50 up to 88.6%, and 1 in 20 (large green dot) up to 95.24%. **Abbreviations:** HMDx, home molecular diagnostic; LAMP, loop-mediated isothermal amplification; NPA, negative percent agreement; PB, prevalence boundary; RAgT, rapid antigen test; R_FO_, rate of false omissions; and WHO, World Health Organization.

**Figure 2 diagnostics-13-03223-f002:**
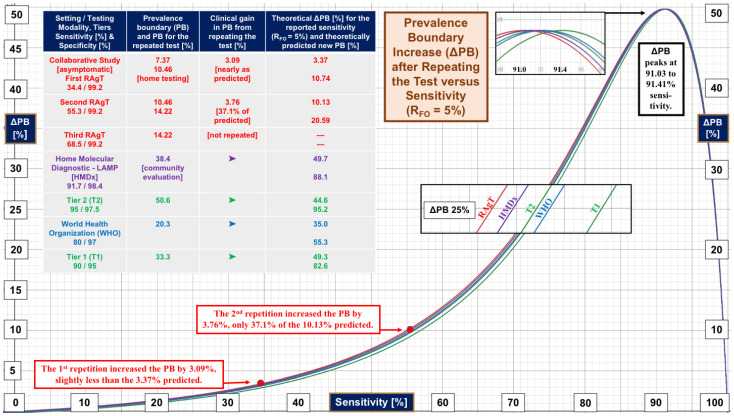
**Gain in Prevalence Boundary as a Function of Test Sensitivity**. This figure illustrates three key findings: (1) The curves (color coded to the inset table) cluster together because of the narrow range in clinical specificity (95% to 99.2%), which means that the primary driver of the increase in prevalence boundary (∆PB) is sensitivity; (2) the shallow shape of the curves on the left emphasizes how little is gained by repeating RAgTs tests that start with very low sensitivity; and (3) when sensitivity is 91.0–91.4%, a repeated test will maximally increase the prevalence boundary as shown by the peaks on the right, making the higher performance tests more useful in settings of different prevalence because missed diagnoses are minimized. Please see the inset table for performance metrics. The curves were created using Equation (26). **Abbreviations:** ∆PB, the increase in PB with repeated testing; PB, prevalence boundary; RAgT, rapid antigen test; R_FO_, rate of false omissions; T1, tier 1, T2, tier 2; and WHO, World Health Organization.

**Figure 3 diagnostics-13-03223-f003:**
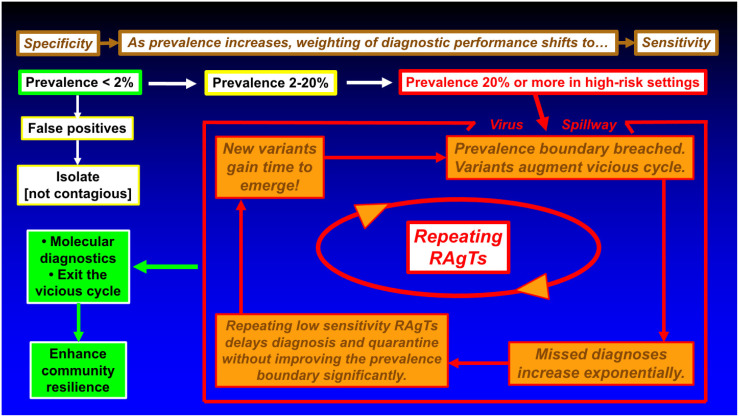
**Potential Vicious Cycle Fueled by Repeating Poorly Performing Rapid Antigen Tests.** Poorly performing RAgTs can perpetuate virus transmission by missing diagnoses, more so as prevalence increases and the weighting of test performance shifts from specificity (**top left**) to sensitivity (**top right**). In high-risk settings and hotspots, prevalence breaches and evolving variants may compound an outbreak to generate an epidemic. Repeating RAgTs consumes valuable time. Asymptomatic people may unknowingly spread disease to family, friends, workers, and clients, thereby creating a vicious cycle. **Abbreviation:** RAgTs, rapid antigen tests.

**Table 1 diagnostics-13-03223-t001:** Diagnostic Performance Tiers with Systematic Prevalence Boundaries for Repeated Tests.

Tier	Performance Level	Sensitivity [%]	Specificity [%]	Prevalence Boundary for R_FO_ of 5%		
				1st Test [%]	2nd Test [%]	∆PB [%]
1	Low	90	95	33.3	82.6	49.3
2	Marginal	95	97.5	50.6	95.2	44.6
3	High	100	≥99	No Boundary	No Boundary	—

Abbreviations: PB, prevalence boundary; ∆PB, the gain in PB from repeating the test; and R_FO_, the rate of false omissions (missed diagnoses).

**Table 2 diagnostics-13-03223-t002:** Fundamental Definitions, Derived Equations, Ratios, Rates, Predictive Value Geometric Mean-Squared, Prevalence Boundary, Recursion, and Special Cases.

Eq. No.	Category and Equations	Dep. Var.	Indep. Var.
Fundamental Definitions			
(1)	x = Sens = TP/(TP + FN)	x	TP, FN
(2)	y = Spec = TN/(TN + FP)	y	TN, FP
(3)	s = PPV = TP/(TP + FP)	s	TP, FP
(4)	t = NPV = TN/(TN + FN)	t	TN, FN
(5)	p = Prev = (TP + FN)/N	p	TP, FN, N
(6)	N = TP + FP + TN + FN	N	TP, FP, TN, FN
**Derived Equations**			
(7)	PPV = [Sens·Prev]/[Sens·Prev + (1 − Spec)(1 − Prev)], or s = [xp]/[xp + (1 − y)(1 − p)]—symbolic version of the equation above	s	x, y, p
(8)	p = [s(y − 1)]/[s(x + y − 1) − x]	p	x, y, s
(9)	x = [s(p − 1)(y − 1)]/[p(s − 1)]	x	y, p, s
(10)	y = [sp(x − 1) + s − px]/[s(1 − p)]	y	x, p, s
(11)	NPV = [Spec·(1 − Prev)]/[Prev·(1 − Sens) + Spec·(1 − Prev)], or t = [y(1 − p)]/[p(1 − x) + y(1 − p)]	t	x, y, p
(12)	p = [y(1 − t)]/[t(1 − x − y) + y]	p	x, y, t
(13)	x = [pt + y(1 − p)(t − 1)]/[pt]	x	y, p, t
(14)	y = [pt(x − 1)]/[t(1 − p) − 1 + p]	y	x, p, t
**Ratios**			
(15)	TP/FP = PPV/(1 − PPV) = [Sens·Prev]/[(1 − Spec)(1 − Prev)], or [xp]/[(1 − y)(1 − p)]	TP/FP Ratio	x, y, p
(16)	FP/TP = (1 − PPV)/PPV = [(1 − y)(1 − p)]/(xp)	FP/TP Ratio	x, y, p
(17)	FN/TN = (1 − NPV)/NPV = [p(1 − x)]/[y(1 − p)]	FN/TN Ratio	x, y, p
**Rates**			
	*True positive* (*R_TP_*), *false positive* (*R_FP_*), and *positive* (*R_POS_*)		
(18)	R_TP_ = TP/(TP + FN) = x	R_TP_	TP, FN
(19)	R_FP_ = FP/(TN + FP) = 1 − Spec = 1 − y	R_FP_	TN, FP
(20)	R_POS_ = (TP + FP)/N	R_POS_	TP, FP, N
	*False Omission* (*R_FO_*)		
(21)	R_FO_ = FN/(TN + FN) = 1 − NPV = 1 − t = [p(1 − x)]/[p(1 − x) + y(1 − p)]	R_FO_	x, y, p
	*R_FO_ with repeated test* (*rt*)		
(22)	R_FO/rt_ = [p(1 − x)^2^]/[p(1 − x)^2^ + y^2^(1 − p)]	R_FO/rt_	x, y, p
**Predictive value geometric mean-squared (range 0 to 1)**			
(23)	PV GM^2^ = PPV·NPV = s·t = {[xp]/[xp + (1 − y)(1 − p)]}· {[y(1 − p)]/[p(1 − x) + y(1 − p)]}	PV GM^2^	x, y, p
**Prevalence Boundary**			
	*Prevalence boundary for one test given R_FO_*		
(24)	PB = y(1 − t)/[(1 − x) − (1 − t)(1 – x − y)] = [yR_FO_]/[(1 − x) − R_FO_(1 – x − y)] = [yR_FO_]/[R_FO_(x + y − 1) + (1 − x)]	PB	x, y, t or x, y, R_FO_
	*Prevalence boundary for repeated test* (*PB_rt_*) *given R_FO_*		
(25)	PB_rt_ = [y^2^R_FO_]/[R_FO_(y^2^ − x^2^+2x − 1) + (x − 1)^2^]	PB_rt_	x, y, R_FO_
	*Improvement in prevalence boundary* (*∆PB*) *when test second time given R_FO_*		
(26)	∆PB = {y^2^R_FO_/[R_FO_(y^2^ − x^2^ + 2x − 1) + (x − 1)^2^]} − {yR_FO_/[R_FO_/[(x + y − 1) + (1 − x)]}	∆PB	x, y, R_FO_
**Recursion**			
	*Recursive formulae for PPV* (*s_i_*_+1_) *and NPV* (*t_i_*_+1_)		
(27)	s_i+1_ = [xp_i_]/[xp_i_ + (1 − y)(1 − p_i_)], where the index, i = 1, 2, 3…	s_i+1_	x, y, p_i_
(28)	t_i+1_ = [y(1 − p_i_)]/[p_i_(1 − x) + y(1 − p_i_)]	t_i+1_	x, y, p_i_
**Special Cases**			
	*PPV when sensitivity is 100%*		
(29)	PPV = [Prev]/[Prev + (1 − Spec)·(1 − Prev)], or s = [p]/[p + (1 − y)(1 − p)]	s	y, p
	*Prevalence when sensitivity is 100%* (i.e., *FN* = 0)		
(30)	Prev = 1 − [(1 − N_+_/N)/Spec], or p = 1 − [(1 − POS_%_)/y]	p	POS_%_, y
	*Sensitivity when given specificity, R_FO_, and PB* (*no repeat*)		
(31)	x = [PB − R_FO_(y + PB − y·PB)]/[PB(1 − R_FO_)]	x	y, R_FO_, PB
	*Sensitivity, given R_FO_ and PB, when specificity (y) is 100%*		
(32)	x = (PB − R_FO_)/[PB(1 − R_FO_)]	x	R_FO_, PB
	*Accuracy (not recommended—see note)*		
(33)	A = (TP + TN)/N = Sens·Prev(dz) + Spec·Prev(no dz)	A	TP, TN, N

**Abbreviations**: Dep. Var., dependent variable; Eq., equation; FN, false negative; FP, false positive; i, an index from 1 to 3 or more (the number of testing events); Indep. Var., independent variable(s); N, total number of people tested; N_+_, number of positives (TP + FP) in the tested population; NPV, negative predictive value (t); p_i+1_, p_i_, indexed partition prevalence in the recursive formula for PPV and NPV; PB, prevalence boundary; PB_rt_, prevalence boundary for repeated test; ∆PB, improvement in prevalence boundary; POS_%_, (N_+_/N), percent positive of the total number tested (same as R_POS_); PPV, positive predictive value (s); Prev, prevalence (p); Prev(dz), same as p; Prev(no dz), prevalence of no disease; PV GM^2^, square of the geometric mean of positive and negative predictive values, (PPV·NPV), expressed as a fraction from 0 to 1; R_FO_, the rate of false omissions; R_FO/rt_, rate of false omission with repeated test (rt); R_FP_, false positive rate, aka false positive alarm (probability that a false alarm will be raised or that a false result will be reported when the true value is negative); R_POS_, positivity rate; R_TP_, true positive rate, the same as sensitivity; Sens, sensitivity (x); Spec, specificity (y); TN, true negative; and TP, true positive. **Notes**: Sens, Spec, PPV, NPV, and Prev are expressed as percentages from 1 to 100%, or as decimal fractions from 0 to 1 by dividing by 100%. PV GM^2^ was created for visual logistics comparisons of performance curves of diagnostic tests, not for point comparisons. If the denominators of derived equations become indeterminate, then revert to the fundamental definitions, Equations (1)–(6). The use of the formula for accuracy [Equation (33)] is not recommended because of the duplicity of values with complementary changes in sensitivity and specificity.

## Data Availability

Data is available upon reasonable request from the author.

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
