# Peer review of "The Impact of Repeating COVID-19 Rapid Antigen Tests on Prevalence Boundary Performance and Missed Diagnoses"

_diagnostics, 2023, doi:10.3390/diagnostics13203223_

Round 1

Reviewer 1 Report

The  article "THE IMPACT OF REPEATING COVID-19 RAPID ANTIGEN TESTS ON ON PREVALENCE BOUNDARY PERFORMANCE AND MISSED DIAGNOSES "by Gerald J Kost is devoted to formulation of the mathematical basis for COVID-19 rapid antigen tests improvement. This research seems especially relevant and useful in the context of future pandemics. Thus, I find that the topic aligns well with the journal's scope and is relevant to current research interests.

However, I recommend a minor revision to address the following point:

30-35. It is necessary to add the references to the works of other researchers besides the author confirming the statements.

Author Response

Please see the attached PDF, which has both author responses and tracked revisions. Thank you for helping to improve this manuscript.

Reviewer 2 Report

Thank you for sharing your article on the impact of repeated COVID-19 antigen testing. The comments on your manuscript may help to improve it:

L39: Are three tests also to be performed during a time fame of 5 days? Please clarify in your manuscript.

L103: Please provide in-depth information on the participants enrolled and a detailed justification of the sample size chosen.

L143: Add the software versions used.  

General comment: Please include a statement about consent procedures. 

Please see above. 

Author Response

(The authors gave the same response as above.)

Round 2

Reviewer 2 Report

All my comments were addressed sufficiently. 

Please see above.